# Temporal Transcriptomics of Gut *Escherichia coli* in *Caenorhabditis elegans* Models of Aging

Joshua D. Brycki,[a] Jeremy R. Chen See,[a,b] Gillian R. Letson,[a] Cade S. Emlet,[a] Lavinia V. Unverdorben,[a] Nathan S. Heibeck,[a] Colin J. Brislawn,[c] Vincent P. Buonaccorsi,[a] Jason P. Chan,[d] Regina Lamendella[a,b]

[a]Juniata College, Huntingdon, Pennsylvania, USA
[b]Wright Labs LLC, Huntingdon, Pennsylvania, USA
[c]Contamination Source Identification, Huntingdon, Pennsylvania, USA
[d]Marian University, Indianapolis, Indiana, USA

Joshua D. Brycki and Jeremy R. Chen See are co-first authors and contributed equally to this article. Author order was determined alphabetically by last name.

**ABSTRACT** Host-bacterial interactions over the course of aging are understudied due to complexities of the human microbiome and challenges of collecting samples that span a lifetime. To investigate the role of host-microbial interactions in aging, we performed transcriptomics using wild-type *Caenorhabditis elegans* (*N2*) and three long-lived mutants (*daf-2*, *eat-2*, and *asm-3*) fed *Escherichia coli* OP50 and sampled at days 5, 7.5, and 10 of adulthood. We found host age is a better predictor of the *E. coli* expression profiles than host genotype. Specifically, host age was associated with clustering (permutational multivariate analysis of variance [PERMANOVA], $P = 0.001$) and variation (Adonis, $P = 0.001$, $R^2 = 11.5\%$) among *E. coli* expression profiles, whereas host genotype was not (PERMANOVA, $P > 0.05$; Adonis, $P > 0.05$, $R^2 = 5.9\%$). Differential analysis of the *E. coli* transcriptome yielded 22 Kyoto Encyclopedia of Genes and Genomes (KEGG) pathways and 100 KEGG genes enriched when samples were grouped by time point [LDA, linear discriminant analysis; log(LDA), $\geq 2$; $P \leq 0.05$], including several involved in biofilm formation. Coexpression analysis of host and bacterial genes yielded six modules of *C. elegans* genes that were coexpressed with one bacterial regulator gene over time. The three most significant bacterial regulators included genes relating to biofilm formation, lipopolysaccharide production, and thiamine biosynthesis. Age was significantly associated with clustering and variation among transcriptomic samples, supporting the idea that microbes are active and plastic within *C. elegans* throughout life. Coexpression analysis further revealed interactions between *E. coli* and *C. elegans* that occurred over time, building on a growing literature of host-microbial interactions.

**IMPORTANCE** Previous research has reported effects of the microbiome on health span and life span of *Caenorhabditis elegans*, including interactions with evolutionarily conserved pathways in humans. We build on this literature by reporting the gene expression of *Escherichia coli* OP50 in wild-type (*N2*) and three long-lived mutants of *C. elegans*. The manuscript represents the first study, to our knowledge, to perform temporal host-microbial transcriptomics in the model organism *C. elegans*. Understanding changes to the microbial transcriptome over time is an important step toward elucidating host-microbial interactions and their potential relationship to aging. We found that age was significantly associated with clustering and variation among transcriptomic samples, supporting the idea that microbes are active and plastic within *C. elegans* throughout life. Coexpression analysis further revealed interactions between *E. coli* and *C. elegans* that occurred over time, which contributes to our growing knowledge about host-microbial interactions.

**KEYWORDS** *Caenorhabditis elegans*, *Escherichia coli*, aging, bacteria, microbiome, transcriptomics

Address correspondence to Regina Lamendella, lamendella@juniata.edu.com.

Approximately 40% of U.S. adults and 80% of those over the age of 65 have multiple chronic conditions (1). The average American is getting older, and the census bureau projects there will be more adults over the age of 65 than children under 18 by 2034 (2). This marks an important trend for the future of health care, as chronic and mental health conditions already account for 90% of the nation's $3.8 trillion annual health care costs (1, 3). Aging research is at the forefront of mitigating this trend in search of ways to promote healthy aging and extending health span, which is the number of years an individual is healthy and disease free.

Analysis of the microbiome is an emerging area of aging research since it has been shown to affect health span and to change over the course of aging (4–7). In fact, the microbiome affects all nine hallmarks of aging (8, 9), including genomic instability (10), telomere attrition (9), epigenetic alterations (11, 12), loss of proteostasis (13, 14), deregulated nutrient sensing (15, 16), mitochondrial dysfunction (17), cellular senescence (18), stem cell exhaustion (19), and altered intercellular communication (20). Disentangling these effects in humans can be very challenging because of variability in microbiome composition caused by lifestyle (21–23), difficulty collecting samples that span a lifetime, and the complexity of the human microbiome.

Using a *Caenorhabditis elegans* model fed *Escherichia coli* OP50 systematizes the study of host-microbial interactions and aging by minimizing complexity, sampling difficulty, and external variables. The *C. elegans* microbiome has been associated with increased availability of bacterial metabolites like amino acids, nitrous oxide, iron, and folic acid, of which each affects host health span and life span (15, 24–26). Bacteria within *C. elegans* have also been shown to metabolize drugs, such as the cancer therapeutic 5-fluorouracil (27) and the antidiabetic metformin (28) into metabolites that alter host physiology. In reaction to pathogenic bacteria, *C. elegans* produces reactive oxygen species as an immune response, a process necessary for healthful aging amid bacterial stressors (29). These examples in *C. elegans* are just several in a growing list of host-microbial interactions important for aging.

We investigated bacterial gene expression changes over time and sought to understand how those changes could contribute to host aging. To do this investigation, we used host-bacterial transcriptomics, which provides a comprehensive view of host-microbial activity at the level of mRNA. Specifically, we looked at the transcriptome of four *C. elegans* animals (wild-type *N2* and long-lived *daf-2* [30], *eat-2* [31], and *asm-3* [32]) and their resident bacteria (*E. coli* OP50) at days 5, 7.5, and 10 of adulthood. This time frame represents the duration from fully established bacterial colonization through host senescence (33). The mutants selected represent two well-studied mechanisms of life span extension in *C. elegans*, as follows: caloric restriction (*eat-2*) and mutations in insulin/insulin-like growth factor 1 (IGF-1) signaling (*asm-3* and *daf-2*) (34). We found expression changes over time in such genes involved in amino acid metabolism, biofilm formation, and other functions. Observing these changes over time unveiled a previously unknown role of host-microbial interactions in aging research. While host genotype has been shown previously to affect bacterial transcriptomics pertaining to gene presence and absence in a *C. elegans* model (4), to the best of our knowledge, we are the first to report age-dependent host-microbial transcriptomics profiles in the *C. elegans* model organism.

## RESULTS

**Overview of the transcriptomic profile.** In total, 42 samples were sequenced and analyzed, with each sample yielding an average of 16 million sequences. Sequencing data were filtered to remove host RNA, rRNA, and bacterial sequences other than *E. coli* OP50 (Table 1). Of the remaining sequences, those mapping to *C. elegans* ($n = 57,508,699$) and *E. coli* OP50 ($n = 5,013,461$) were used for subsequent analysis.

**Age is a better predictor of the microbial transcriptome than host genotype.** Our experimental design had two independent variables, namely, age (5-, 7.5-, and 10-day adults) and host genotype (*N2*, *daf-2*, *asm-3*, and *eat-2*). We performed an analysis of the bacterial transcriptome based on presence-absence and expression enrichment.

**TABLE 1** RNA Sequencing results of all experimental sample groups

| Experimental group | No. of samples | Total raw reads | Total reads posttrimming | rRNA reads removed | Human reads removed | Other bacterial reads removed | *C. elegans* mapped reads | *E. coli* OP50 mapped reads | Total identified *E. coli* KEGG orthologs |
|---|---|---|---|---|---|---|---|---|---|
| *N2* Day 5 | 3 | 51,368,564 | 48,624,914 | 35,588,703 | 436,254 | 4,838,569 | 4,043,440 | 336,432 | 2,044 |
| *N2* Day 7.5 | 3 | 101,113,426 | 92,869,775 | 46,093,638 | 368,780 | 33,172,149 | 7,238,186 | 337,452 | 2,176 |
| *N2* Day 10 | 3 | 21,180,306 | 19,256,960 | 9,003,190 | 1,342 | 9,022,173 | 489,464 | 76,088 | 2,203 |
| *daf-2* Day 5 | 3 | 94,768,625 | 87,801,144 | 56,068,764 | 187,101 | 13,212,271 | 12,384,743 | 314,527 | 2,250 |
| *daf-2* Day 7.5 | 4 | 54,563,772 | 49,966,566 | 26,890,582 | 923,883 | 15,662,740 | 1,727,284 | 2,625,785 | 1,767 |
| *daf-2* Day 10 | 4 | 41,268,259 | 39,001,308 | 16,890,627 | 128,333 | 16,153,141 | 3,543,722 | 130,296 | 1,726 |
| *eat-2* Day 5 | 3 | 32,086,445 | 30,596,996 | 18,940,926 | 801,437 | 5,532,820 | 3,014,322 | 228,419 | 2,012 |
| *eat-2* Day 7.5 | 3 | 66,328,124 | 62,092,391 | 39,009,811 | 1,072,618 | 7,838,332 | 9,751,033 | 158,052 | 2,123 |
| *eat-2* Day 10 | 4 | 37,241,549 | 34,866,144 | 16,696,371 | 256,920 | 14,757,095 | 1,549,659 | 131,896 | 1,780 |
| *asm-3* Day 5 | 4 | 45,874,617 | 42,901,316 | 32,280,006 | 495,422 | 4,108,678 | 3,072,654 | 268,528 | 1,990 |
| *asm-3* Day 7.5 | 4 | 57,576,223 | 53,295,984 | 28,621,298 | 311,525 | 16,566,012 | 4,089,826 | 188,066 | 1,919 |
| *asm-3* Day 10 | 4 | 76,608,761 | 70,556,191 | 36,855,510 | 359,350 | 22,502,718 | 6,604,366 | 217,920 | 1,928 |

Comparisons across age and genotypes yielded unique transcriptomes based on enrichment comparisons (Table 2) and presence-absence comparisons of the core (>70% of samples in that genotype) transcriptome (Fig. 1). Presence-absence comparisons of the core transcriptome between mutants yielded 195 uniquely expressed genes (Fig. 1A), whereas comparisons across time yielded 256 unique genes (Fig. 1B). Not all of these genes were significantly enriched in expression compared with the other mutants due to low expression levels and high variance, but these results indicate that age had a stronger relationship with our residential bacterial transcriptome than genotype.

Enrichment analysis of the bacterial transcriptome using linear discriminant analysis (LDA) effect size (LEfSe) [log(LDA), $\geq 2$; $P \leq 0.05$] yielded multiple enriched genes and pathways (Table 2; Table S1 to S38 in the supplemental material). Age comparisons of all samples collated yielded many more enriched genes ($n = 100$) and pathways ($n = 22$) than across genotype comparisons ($n = 15$ and 4, respectively). This trend continued after data visualization (nonmetric multidimensional scaling [NMDS] and partial least squares-discriminant analysis [PLS-DA]) and distance matrix analysis (permutational multivariate analysis of variance [PERMANOVA]) of expression data across ages and genotypes. Microbial transcriptomes did not cluster based on genotype (PERMANOVA, $P > 0.05$) using NMDS or PLS-DA analysis (Fig. 2A and B). In contrast, NMDS and PLS-DA with samples grouped by age yielded distinct clustering (PERMANOVA, $P = 0.001$) (Fig. 2C and D). Furthermore, PLS-DA had lower overall error for the model separating samples by time point (see Fig. S1 in the supplemental material) than the model for samples by genotype (see Fig. S2 in the supplemental material). Within each genotype, clustering based on time point was also observed in PLS-DA plots

**TABLE 2** Summary of LEfSe comparisons

| Class | LEfSe comparison | No. of enriched genes or pathways [log(LDA) $\geq 2$, $P \leq 0.05$] |
|---|---|---|
| Genes | Genes across age | 100 |
| | Genes across mutants | 15 |
| | Genes across age, genotypes separated | 290 |
| | Genes with each mutant compared with *N2* | 270 |
| | Genes within each mutant compared with *N2* across time | 885 |
| Pathways | Pathways across age | 22 |
| | Pathways across mutants | 4 |
| | Pathways across age, genotypes separated | 37 |
| | Pathways within each mutant compared with *N2* | 15 |
| | Pathways within each mutant compared with *N2* across time | 81 |

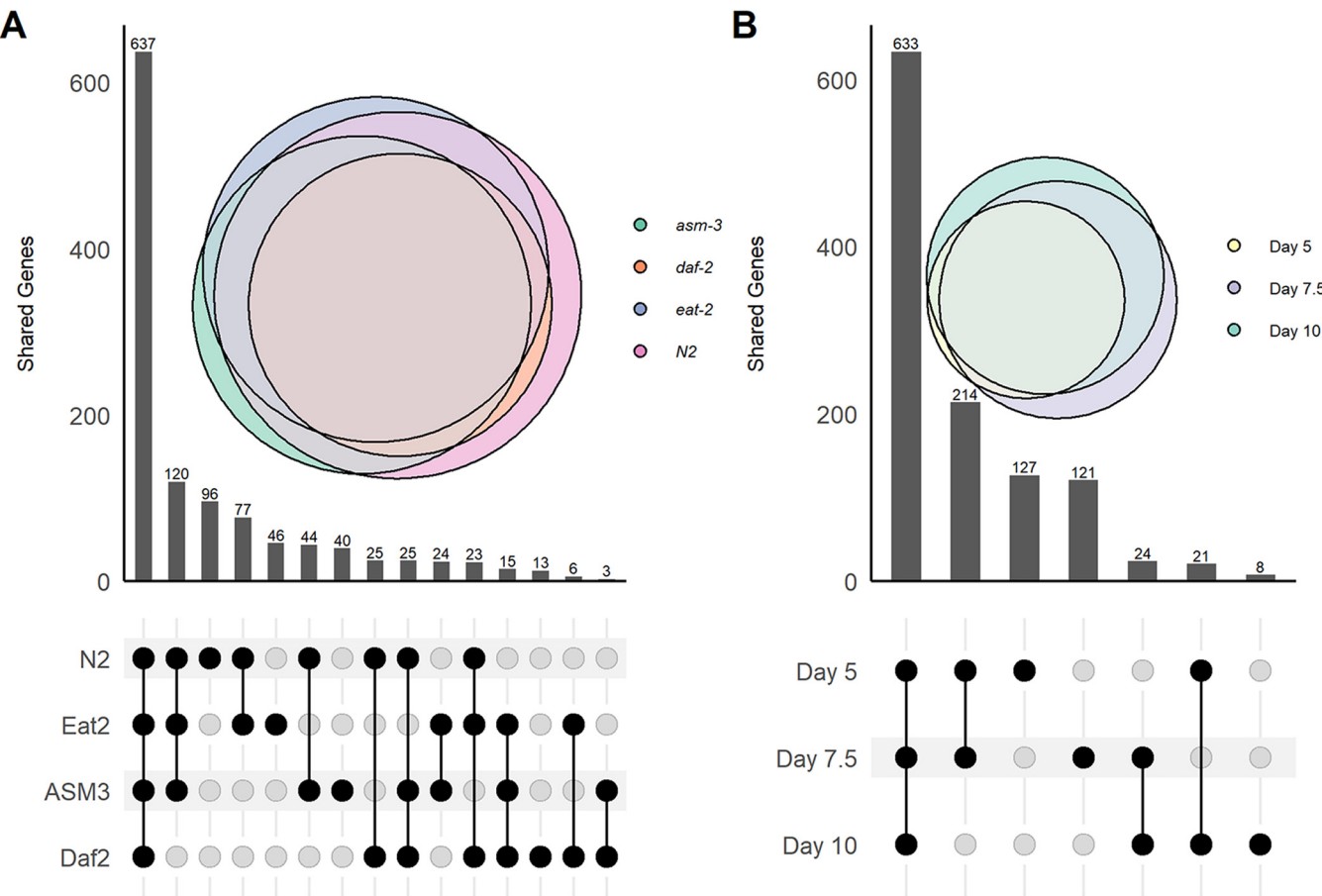

**FIG 1** Overlap comparison of *E. coli* core transcriptomes when samples were grouped by genotype (A) and time point (B). Core transcriptomes were genes (KEGG orthologs) that were expressed in at least 70% of the samples in a group. Overlap was visualized with the eulerr and UpSetR packages in R.

(see Fig. S3 and S6 in the supplemental material), although they had higher error rates than the time point model made with all samples (see Fig. S7 and S10 in the supplemental material). These results provide evidence that the *E. coli* transcriptome was more differentiated according to host age than to host genotype.

While all mutations were treated as separate for the above PLS-DA and PERMANOVA analyses, each mutant was also compared to the wild type (*N2*) for PERMANOVA analysis. This pairwise analysis addresses the possibility that a shared pathway of two mutants, namely, *asm-3* and *daf-2*, led to insignificant differences between mutants when collated. Comparisons between mutants and *N2* were not statistically significant (Table 3). In contrast, each time point was statistically significantly different (Table 3). These results support the idea that age has a larger impact on the microbial transcriptome in our study than host genotype.

**Microbial functional expression changes over time.** Partitioning variance based on time point using PERMANOVA revealed that time point explained 11.4% of the variation ($P = 0.001$) in *E. coli* gene expression, which is almost twice as much as could be explained by genotype when all samples were used ($R^2 = 0.059$, $P = 0.88$) or in any of the four subsets containing only three genotypes (Table 4). In combination, time point and genotype did explain slightly more variation among samples (13.2%), although not significantly ($P > 0.05$), unlike time point by itself. Furthermore, some of the changes in expression associated with time point have potential implications for host health. For example, biofilm formation and membrane transport pathways were identified as differentially expressed when samples were grouped by time point (Fig. 3). Four amino acid biosynthesis and utilization pathways were also identified as differentially expressed in that comparison, with three of them having at least one individual gene

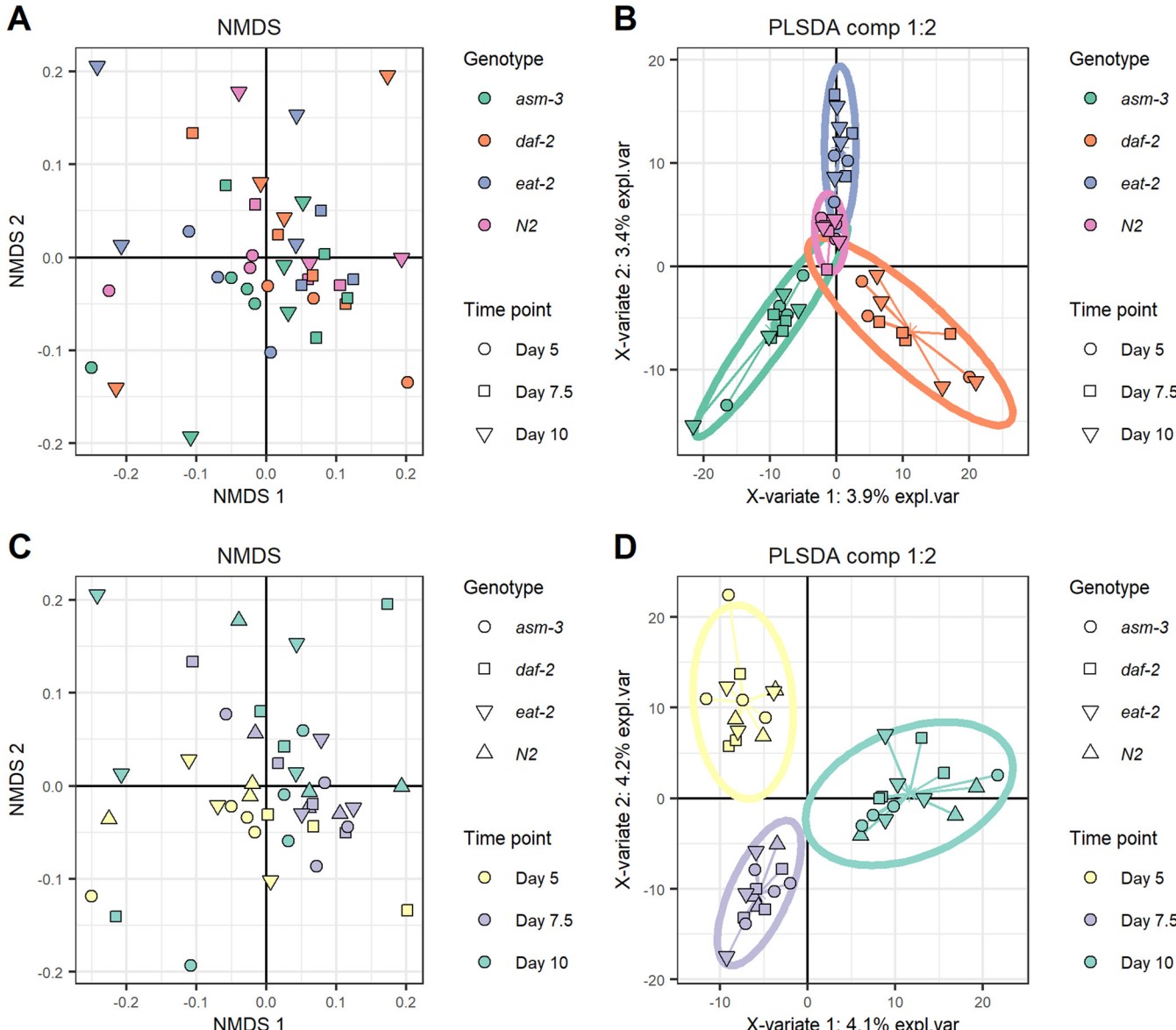

**FIG 2** Nonmetric multidimensional scaling and partial least squares discriminant analysis (PLS-DA) plots. (A) NMDS plot of samples colored by genotype. (B) PLS-DA plot of samples grouped by genotype. (C) NMDS plot of samples colored by time point. (D) PLS-DA plot of samples grouped by time point. NMDS and PLS-DA analyses were conducted through R using the phyloseq and mixOmics packages with a CPM-normalized table of the KEGG orthologs that were attributed to *E. coli*. Ellipses around the groups indicate 95% confidence intervals, meaning a lack of overlap between cohorts indicates a defined gene expression profile among them.

that was also significantly enriched based on time point (Table S19 and S38). Within the day 10 time point as well, when mutants were compared to the wild type, glutathione metabolism (ko00480) was found to be significantly more expressed by *E. coli* in the *asm-3* and *daf-2* mutants than by *N2*. Pathway networks were created for this pathway, as well as the three differential biofilm pathways for the time point comparison. These networks revealed multiple genes were expressed within each pathway, contributing to their enrichment (Fig. S11 and S22 in the supplemental material).

**Microbial genes coexpressed with host genes.** Changes in bacterial gene expression could influence host gene expression, including genes related to aging. Understanding this relationship may provide insight into how gut bacteria affect longevity. To investigate this possibility, we used the program Lemon-Tree to predict interactions between hosts and residential bacteria by finding bacterial regulators of *C. elegans* gene expression. This analysis revealed six modules of *C. elegans* genes that

**TABLE 3** Summary of PERMANOVA comparisons

| Group 1 | Group 2 | Sample size | Permutations | pseudo-F | P value | q value |
|---|---|---|---|---|---|---|
| Day 5 | Day 10 | 28 | 999 | 2.29852191 | 0.003 | 0.0045 |
| Day 5 | Day 7.5 | 27 | 999 | 3.466195887 | 0.002 | 0.0045 |
| Day 7.5 | Day 10 | 29 | 999 | 2.024045542 | 0.027 | 0.027 |
| asm-3 | daf-2 | 23 | 999 | 1.023993612 | 0.393 | 0.924 |
| asm-3 | eat-2 | 22 | 999 | 0.970479004 | 0.443 | 0.924 |
| asm-3 | N2 | 21 | 999 | 0.715339934 | 0.857 | 0.924 |
| daf-2 | eat-2 | 21 | 999 | 0.771789704 | 0.767 | 0.924 |
| daf-2 | N2 | 20 | 999 | 0.662995617 | 0.924 | 0.924 |
| eat-2 | N2 | 19 | 999 | 0.615302894 | 0.897 | 0.924 |
| asm-3_5 | asm-3_10 | 8 | 999 | 1.580322208 | 0.102 | 0.418 |
| asm-3_5 | asm-3_7.5 | 8 | 999 | 2.15662942 | 0.037 | 0.418 |
| asm-3_7.5 | asm-3_10 | 8 | 999 | 0.842597992 | 0.632 | 0.83424 |
| daf-2_5 | daf-2_10 | 7 | 999 | 1.315018691 | 0.166 | 0.498 |
| daf-2_5 | daf-2_7.5 | 7 | 999 | 0.964443897 | 0.484 | 0.763714286 |
| daf-2_7.5 | daf-2_10 | 8 | 999 | 0.829183353 | 0.687 | 0.855509434 |
| eat-2_5 | eat-2_10 | 7 | 999 | 1.049002005 | 0.486 | 0.763714286 |
| eat-2_5 | eat-2_7.5 | 6 | 999 | 1.834255897 | 0.093 | 0.418 |
| eat-2_7.5 | eat-2_10 | 7 | 999 | 1.520096102 | 0.114 | 0.418 |
| N2_5 | N2_10 | 6 | 999 | 1.673133646 | 0.094 | 0.418 |
| N2_5 | N2_7.5 | 6 | 999 | 1.657505126 | 0.09 | 0.418 |
| N2_7.5 | N2_10 | 6 | 999 | 0.842056733 | 0.61 | 0.83424 |

each had a bacterial regulator (Table 5; see Table S39 in the supplemental material). The regulators with the strongest change in expression over time were diguanylate cyclase (DgcP) and phosphomethylpyrimidine synthase (ThiC), which had slopes of −0.216 and −0.173, respectively. The regulator UDP-3-O-(3-hydroxymyristoyl) glucosamine N-acyltransferase (LpxD) had a slope of −0.050. DgcP and LpxD pertain to extracellular expression of the bacteria, being involved with biofilm formation and lipopolysaccharide (LPS) production, respectively. ThiC is involved in thiamine production. Notably, the only two modules with a positive slope pertained to bacterial ribosomes (Table 5).

**DAF-16 transcriptional regulation in *C. elegans*.** When analyzing bacterial coexpressional data, we observed an abundance of insulin/IGF-1 signaling pathway (IIS)-regulated genes with differential expression based on time point. In fact, 80 out of 431 *C. elegans* genes present in a coexpressed gene module are known to be regulated by DAF-16, the best studied transcription factor in the IIS pathway and which acts opposite the factor PQM-1 (see Table 40 in the supplemental material) (35). Of these 80 genes, 37 are known to be upregulated (class 1) and 43 are known to be downregulated (class 2) by DAF-16 nuclear localization. Our observed class 1 and class 2 genes did not separate into different regulatory modules and were present in both positively and negatively sloping modules. Specifically, 36 class 1 genes were in modules with negative slopes, while 1 was in a module with a positive slope. Likewise, 33 class 2 genes were in modules with negative slopes, and 10 were in modules with positive slopes.

**TABLE 4** Summary of Adonis comparisons based on *E. coli* expression using the Bray-Curtis distance metric[a]

| Dataset | Variable | Df[b] | SumsOfSqs[b] | MeanSqs[b] | F.Model[b] | R2 | Pr(>F)[b] |
|---|---|---|---|---|---|---|---|
| Full | TP | 2 | 0.277480486 | 0.138740243 | 2.51984881 | 0.114435337 | 0.001 |
| Full | TP:genotype | 6 | 0.319590966 | 0.053265161 | 0.946338246 | 0.13180206 | 0.61 |
| Full | Genotype | 3 | 0.14416488 | 0.04805496 | 0.800700071 | 0.059454835 | 0.88 |
| No_N2 | Genotype | 2 | 0.112309675 | 0.056154837 | 0.920731037 | 0.057832208 | 0.595 |
| No_eat-2 | Genotype | 2 | 0.093520559 | 0.04676028 | 0.810190806 | 0.0529184 | 0.814 |
| No_daf-2 | Genotype | 2 | 0.09112913 | 0.045564565 | 0.77504047 | 0.052456064 | 0.845 |
| No_asm-3 | Genotype | 2 | 0.086198993 | 0.043099496 | 0.687633722 | 0.048467118 | 0.949 |

[a]All samples were included in the "full" data set comparisons, while samples indicated were omitted from the "no-" comparisons.
[b]Df, degrees of freedom; SumsOfSqs, sums of squares; MeanSqs, mean of squares; F.Model, pseudo-F statistic from model; Pr(>), probability of observing a larger pseudo-F statistic by chance.

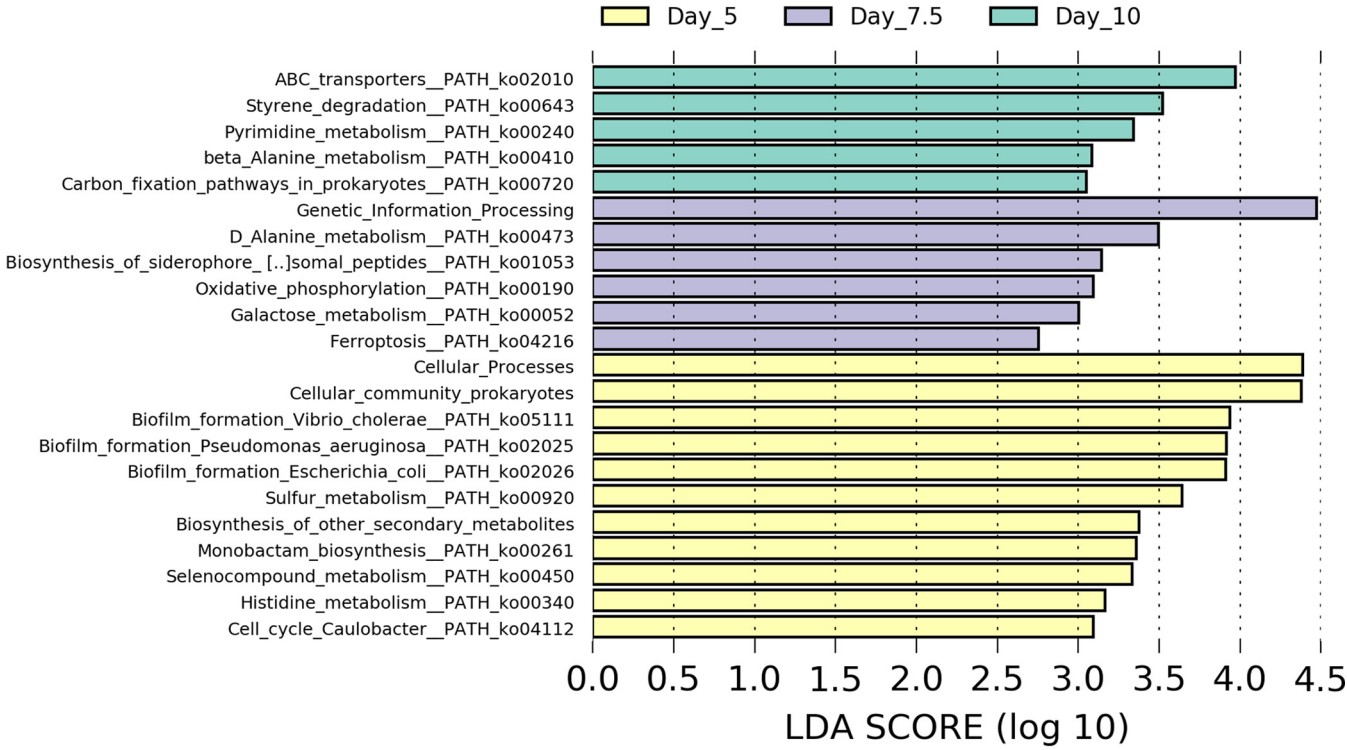

**FIG 3** LEfSe analysis of expressed *E. coli* pathways for all samples grouped by time point. Differential genes were identified through LEfSe using Kruskal-Wallis tests ($P \leq 0.05$), and enrichment was qualified using linear discriminant analysis, with a log(LDA) threshold of 2.0 for a pathway to be considered enriched.

## DISCUSSION

This study investigated expression changes of *C. elegans* and intestinal *E. coli* transcriptomes, both over time and in different host genotypes. In addition, we sought to investigate the influence of host-microbial interactions using coexpression analysis of *C. elegans* and *E. coli* transcriptomes. By using multiple long-lived mutants, we were able to investigate if their increased life span was at least partly attributable to bacterial activity in any or all of the mutants. Although our analysis revealed multiple differences among samples based on both time point and genotype, differences according to time point were more consistent and tended to be of a greater magnitude than those associated with genotype.

However, one notable difference between genotypes was the increased expression of bacterial glutathione metabolism (ko00480) at day 10 in the long-lived *asm-3* [log (LDA) = 2.808, *P* = 0.034] and *daf-2* [log(LDA) = 3.001, *P* = 0.034] mutants. This enrichment could be due to increased host immunity compared with the wild type, as glutathione is an important antioxidant that helps protect cells from stress (36). However, *E. coli* can produce and excrete glutathione (37), and the glutathione/glutathione disulfide

**TABLE 5** *C. elegans* gene modules regulated by *E. coli*

| *E. coli* regulator | GO terms coexpressed | *C. elegans* genes coexpressed | Slope of regulator over time |
|---|---|---|---|
| Phosphomethylpyrimidine synthase ThiC | 15 | 67 | −0.173 |
| Diguanylate cyclase DgcP | 5 | 69 | −0.216 |
| UDP-3-*O*-(3-hydroxymyristoyl)glucosamine *N*-acyltransferase LpxD | 7 | 149 | −0.050 |
| 30S Ribosomal protein S19 | 10 | 44 | 0.077 |
| Glycosyltransferase | 1 | 24 | −0.020 |
| Small ribosomal subunit biogenesis GTPase RsgA | 2 | 25 | 0.072 |

couple in *C. elegans* is important for maintaining mucosal integrity in the midst of oxidative stress (38, 39), which increases as worms age (40, 41). Furthermore, some studies have been done on the benefits of administering probiotic bacteria to reduce host oxidative stress (36), with one study explicitly tying *Lactobacillus fermentum* glutathione synthesis and release to decreased colonic inflammation in a rat model (42). Therefore, future work could investigate the possibility of glutathione being both synthesized and released in residential *E. coli* within those two mutants and, if so, to what extent that contributes to their increased life spans.

*C. elegans* also responded to bacteria. Lemon-Tree analysis revealed six bacterial genes which may have regulated host expression (Table 5). One of these regulators, DgcP is known to induce biofilm formation (43–45). This could be the case in our samples, as both this gene and biofilm pathways [ko02026, log(LDA) = 4.077, $P$ = 0.021; ko02025, log(LDA) = 4.085; $P$ = 0.026; and ko05111, log(LDA) = 4.051, $P$ = 0.031] were less expressed at time points day 7.5 and day 10. Therefore, *C. elegans* genes identified as regulated by DgcP likely became active in the host in response to *E. coli* biofilm formation caused by DgcP, in concordance with a previous study that examined *C. elegans* gene expression and biofilm formation (15). Interestingly, that study found that biofilm formation was associated with increased *C. elegans* longevity (15), indicating biofilm formation is not always deleterious. However, different host genes were associated with biofilm in this study, possibly due to using different bacteria or *C. elegans* genotypes. Still, in combination, our study, along with that previous one, suggests biofilm formation can induce changes in *C. elegans* host cell expression (that are not meant to defend against those bacteria).

Another regulator, namely, LpxD, is involved in bacterial lipopolysaccharide (LPS) biosynthesis. LPS is a component of the Gram-negative bacterial cell wall that contributes to bacterial pathogenicity (46), and *C. elegans* may modulate its interaction with pathogenic bacteria through neuronal responses mediated by lipopolysaccharide structure (47–49). The gene module regulated by LpxD includes gene ontology (GO) terms, such as nitrogen response, phosphorylated nucleoside metabolism, amide transport, nucleoside binding, IRE1-mediated unfolded protein response, response to topologically incorrect proteins, and organic acid metabolism (Table S39). LPS acts in the host as a pathogen-associated molecular pattern (PAMP), which activates an innate immune response via the p38 MAPK pathway and downstream immune response genes (50). Immune function of the p38 MAPK pathway has been shown to decline through day 15 of adulthood in a process known as immunosenescence (51). Although we saw this negative temporal expression pattern with LpxD expression, we saw no GO terms clearly pertaining to immune response coregulated in this module, possibly due to the relatively mild pathogenicity of *E. coli*. Instead, LPS in nonpathogenic bacteria may be modulating host responses of the GO terms we identified.

Another group of *C. elegans* genes was identified as being regulated by phosphomethylpyrimidine synthase (ThiC) expressed by *E. coli*. ThiC deals with thiamine biosynthesis, which has been found to affect gene transcription and glucose metabolism in mammals (52, 53). None of the *C. elegans* genes in this study are involved with glucose metabolism or oxidation, but three genes are associated with responding to stimuli (GO:0007635, GO:0009607, and GO:0044419). Additionally, several of the GO terms are associated with the host's immune response, namely, GO:0002376, GO:0004222, and GO:0006952. Taken together, these terms indicate that our *C. elegans* was responding to *E. coli* thiamine biosynthesis and expressing an immune response as a result. Commensal bacteria are known to improve the host's immune system (54), and this module seems to provide one mechanism for that improvement, i.e., priming the immune system, making it more ready to combat any future infections.

*C. elegans* class 1 and class 2 IIS genes were present in all of our Lemon-Tree coregulatory modules (Table S40). Class 1 genes are upregulated, and class 2 genes are downregulated with DAF-16 nuclear localization, as class 1 genes are transcribed by DAF-16 and class two genes by the DAF-16 anticorrelated factor PQM-1 (35). Coexpression

of these classes indicates the presence of additional factors, such as SMK-1 and HSF-1 which have been shown to govern a specific subset of IIS genes, thereby allowing both class 1 and class 2 genes to be coexpressed (55). In fact, transcription factors such as ELT-7, NHR-28, and LEC-8 have known affinity for the DAF-16 associated element (DAE), and each of these transcription factors were found to be highly coexpressed within different regulatory modules in our coexpression analysis. Of the 80 class 1 and 2 genes present in our coregulatory modules, though, only 42 had the DAF-16 binding element (DBE) (GTAAACA or TGTTTAC) and/or the DAE (TGATAAG or CTTATCA) within 2,000 bp upstream of the transcription start site of their first transcript (Table S40) (35). Furthermore, transcription factors such as ELT-7, NHR-28, and LEC-8 have known affinity toward DAE, and each of these transcription factors were found to be highly coexpressed within different regulatory modules in our coexpression analysis. These results indicate that the expression of class 1 and class 2 genes is dependent on several factors in addition to DAF-16 and may allow for a stimuli-specific transcriptomic response. It is notable that some gene modules did have unique expressions of class 1 or 2 genes. Specifically, module 75 was composed primarily of class 2 genes, indicating that the positive expression of this module over time is associated with upregulation of class 2 genes (Table S40).

In conclusion, this is the first study to our knowledge to perform temporal transcriptomic analysis to investigate both *C. elegans* and its associated bacterial gene expression, allowing us to observe temporal changes and interactions between them. Notably, *E. coli* expression changed more over time than by host genotype, indicating that host age impacted the residential bacteria more than physiological changes due to genotype. Still, a comparative analysis with LEfSe revealed differences in *E. coli* expression associated with genotype, of which some could have impacted host health. Furthermore, Lemon-Tree analysis revealed that several specific bacterial genes may have more directly regulated host expression. Future work could focus on these potential regulators and their effect on *C. elegans* expression, as well as on potential benefits to host health due to bacterial expression, such as the biosynthesis of glutathione and histidine. Additionally, methods development to increase the microbial signal relative to host would be invaluable for confirming the trends and findings in this study, as well as for any other work attempting to study *C. elegans* and its microbiome with shotgun sequencing. Lastly, future work should measure the bacterial load within the *C. elegans* to see potential changes over time or with respect to genotype. Overall, despite the many remaining unknowns, this study represents a valuable contribution to the growing work on the influence of the microbiome on host health and aging, as it provides strong evidence of bacterial regulation of host genes over time.

## MATERIALS AND METHODS

**Strains and growth conditions.** *C. elegans* strains *N2*, CB1370 [*daf-2*(e1370) III], DA1116 *eat-2* (ad1116) II, and RB1487 *asm-3*(ok1744) IV were provided by the Caenorhabditis Genetics Center (CGC) (NIH Office of Research Infrastructure Programs [P40 OD010440]). CB1370 was grown at 15°C, whereas all other strains were grown at 20°C unless otherwise noted. Worms were grown on nematode growth media (NGM; 0.25% peptone, 51 mM NaCl, 25 mM [KPO$_4$], 5 $\mu$g/ml cholesterol, 1 mM CaCl$_2$, 1 mM MgCl$_2$, and 2% agar). Worms were synchronized by supplementing NGM plates with 50 $\mu$M 5-fluoro-2′-deoxy-uridine (FUdR) to prevent eggs from hatching.

*E. coli* OP50 was procured from CGC, grown to an optical density of 0.6 in Luria Broth (US Biological), and seeded onto NGM plates with 360 $\mu$l of OP50 each. All plates were prepared according to Wormbook (56) and incubated for an additional 24 h at 20°C before worms were inoculated.

**Transcriptomics sample preparation.** Replicates of 12 age-synchronized plates for each worm strain were prepared as follows. Gravid worms were allowed to grow one generation and subjected to a bleaching protocol, leaving age-synchronized eggs (56). A total of 150 to 200 of these eggs were plated onto NGM plates previously seeded with a standardized lawn of OP50. After 2 days, worms were transferred from NGM to NGM + FUdR plates using M9 and an electronic pipettor. Worms were thereafter transferred to fresh NGM + FUdR plates no less than every 3 days. Worms were collected at 5, 7.5, and 10 days after the first transfer onto NGM + FUdR plates as follows. Worms were washed off plates with M9 into a 15-ml conical tube and centrifuged for 1 minute at 450 × *g*, and the supernatant was aspirated off, leaving the worm pellet intact. The pellet was washed with 10 ml M9 with 100 $\mu$g/ml gentamicin and centrifuged for 1 minute at 200 RPM once again, and the supernatant was aspirated and saved, leaving the worm pellet intact. The wash and centrifuge steps were repeated until 10 washes were completed. An 11th wash was performed with regular M9. A total of 100 $\mu$l of each supernatant as well as

several worms from the last wash of each sample were plated onto LB agar plates and incubated at 37°C. Plates were then monitored for the efficacy of washes on depleting nonintestinal bacterial loads compared with a plate with several worms, which allowed some intestinal bacteria to grow. After the washing steps were completed, the pellet was resuspended in 1 ml of M9 and flash frozen using liquid nitrogen.

**RNA extraction, library preparation, and sequencing.** RNA was extracted from all samples using the Quick-RNA miniprep plus kit (Zymo, Irvine, CA). Before extraction, samples were thawed on ice and then centrifuged at maximum speed (21,130 × $g$) at 10°C for 7 minutes to pellet the worms. As much M9 (~800 to 900 $\mu$l) as possible was removed from each sample, and 600 $\mu$l of cold 1× DNA/RNA Shield (Zymo) was then added to each sample and mixed with the pellet by pipetting. The entire volume was added to a lysis tube provided by the kit. Samples were lysed for 2 minutes using a Disruptor Genie (Scientific Industries, Inc, Bohemia, NY) and then spun down for 30 seconds at 13,000 × $g$. The solid tissues and blood cells and subsequent RNA purification protocol was then followed as specified in the Zymo Quick-RNA miniprep plus kit manual. RNA was eluted using 50 $\mu$l of DNase/RNase-free water instead of 100 $\mu$l to increase concentration and then quantified using a Qubit 4 fluorometer (ThermoFisher Scientific, Waltham, MA) and a 1× double-stranded DNA (dsDNA) high-sensitivity (HS) kit (ThermoFisher Scientific). RNA extracts were stored at –80°C until further processing.

Library preparation of all RNA samples was performed using the Trio RNA-Seq kit (Tecan Genomics, Redwood City, CA) per the manufacturer's instructions. All libraries were quality checked using the 2100 bioanalyzer and the high-sensitivity DNA kit (Agilent Technologies, Santa Clara, CA). The libraries were combined into two separate pools in equimolar concentration and gel purified on a 2% agarose gel using a gel purification kit (Qiagen, Frederick, MD). After purification, the libraries were quality checked a second time using the 2100 bioanalyzer and the high-sensitivity DNA kit (Agilent Technologies) and then shipped on dry ice to the DNA Technologies Core at University of California-Davis for sequencing. Each library was sequenced on one lane of an Illumina HiSeq 4000 instrument with paired 150-bp reads.

**Bioinformatics.** The data quality was assessed using FastQC (57), generating average Q score reports across all sequence files. Trimming was completed with the program fastp (58) using a sliding window filter of 4 bp with a minimum average quality of 28 and a minimum base quality of 15. Reads trimmed to fewer than 90 bp were discarded. Filtered reads were then processed in KneadData (59) to remove human contaminant reads, rRNA, and *C. elegans* contaminant reads. A custom Bowtie 2 (60) database was created using a publicly available *C. elegans* assembly (WBcel235) (61) and mapped against to remove *C. elegans*. Ribosomal *E. coli* genes were obtained from NCBI, and *C. elegans* rRNA genes were obtained from Ensembl Biomart. Reads matching the *E. coli* OP50 genome (GenBank accession number GCA_004355015.1, ASM435501v1) and *C. elegans* assembly (WBcel235) were separated out, and the paired orientation of resulting reads was restored using SeqKit (62). HUMAnN2 (63) was used to identify functional *E. coli* genes and pathways using the Uniref90 database (64).

Functional genes that mapped to *E. coli* were used for downstream analyses and were regrouped as Kyoto Encyclopedia of Genes and Genomes (KEGG) orthology (KO) terms (65). The core expression profiles of time points and genotypes were determined using the QIIME1 "core_microbiome.py" script (66) to select only genes that were expressed in at least 70% of samples within the group. Those core profiles were processed with Tidyverse (67) and then visualized in R with the eulerr (68) and UpSetR (69) packages. *E. coli* annotations underwent counts per million (CPM) normalization prior to LEfSe (70), NMDS, and PLS-DA analysis. The distance matrices for NMDS, PERMANOVA, and Adonis analysis were calculated through QIIME2 with the Bray-Curtis distance metric. Adonis was performed through QIIME2 to see how much variation could be explained by genotype (all four and four subsets where each genotype was omitted from a single subset) and time point, as well as genotype and time point together. The subset Adonis analysis was performed to account for the different number of genotypes and time points being a potential confounding factor when comparing the amount of variation those two groupings explained. Pairwise PERMANOVA tests were also performed through QIIME2. In total, 18 LEfSe comparisons were performed for genes (Table S2 to S19). KOs were then regrouped into pathways using a custom Python script, and the same 18 comparisons were then run with the resulting pathways table after first removing pathways that are associated with eukaryotic organisms, i.e., those in human diseases or organismal systems groups (Table S20 to S37). Pathway networks were visualized for the three differentially expressed biofilm pathways identified by the LEfSe analysis for time point (Fig. 3), as well as the glutathione pathway described in the Discussion using the pathview package in R with the counts per million-normalized KO table. For each pathway, three networks were created, namely, specific, genotype, and time point. Each specific network node was divided into 12 sections, with 1 per group (time point by genotype) ordered as follows: day10_*asm-3*, day10_*daf-2*, day10_*eat-2*, day10_*n2*, day5_*asm-3*, day5_*daf-2*, day5_*eat-2*, day5_*n2*, day7.5_*asm-3*, day7.5_*daf-2*, day7.5_*eat-2*, and day7.5_*N2*. Within the network diagrams for the genotype figures, each node was split into four sections, namely, *asm-3*, *daf-2*, *eat-2*, and *N2*, and each node was split into three sections within the time point networks, including day 10, day 5, and day 7.5. NMDS and PLS-DA analyses were performed using the counts per million-normalized KO table through R using the phyloseq and mixOmics packages, respectively, (71) with samples grouped by genotype and then again with samples grouped by time point.

Lemon-Tree analysis (72, 73) was performed to identify gene coexpression modules and assign regulators to those modules. A total of 3,081 putative *C. elegans* regulator genes with GO terms, including kinase activity, signal transduction, regulation of expression, and regulation of DNA templated transcription, were identified using Ensemble Biomart (74). Counts were CPM normalized using EdgeR (75, 76). All 42 samples were used for clustering. Genes and transcripts were filtered using the EdgeR function "filterByExpr" for having CPM of at least five in at least four samples (minimum group size), resulting in 9,880 *C. elegans* and 1,282 *E. coli* genes. Counts were normalized for different library sizes and variance

of low count genes stabilized using the rlog function of DESeq2 and using the "blind false" option to allow for group-specific corrections to variance (77). Finally, expression levels for each gene and transcript were standardized (subtract mean and divide by standard deviation) using R (78). All *E. coli* genes were also included as potential regulators. This procedure left 8,081 *C. elegans* genes for the generation of coexpressed gene modules. Twelve replicates of the initial "ganesh" cluster were performed. Coexpression modules were filtered for having at least 10 transcript members. Assigned regulators were filtered to include only those that scored in the top 1% of genes for those modules.

In order to identify modules corresponding to patterns of interest in the present study, mean expression over genes for each sample within each of the 137 (genes) modules (tight clusters) was used as a response variable for an analysis of covariance (ANCOVA) (mean standardized rlog module expression of ~MUTANT|DAY) using R. Modules with significant terms (false discovery rate [FDR], ≤0.05) were selected for further analysis. Modules were sorted by magnitude of the difference between groups in mean expression change over time. GO enrichment analysis (FDR, ≤0.05) was then performed within GOrilla (79). The set of nonregulatory *C. elegans* genes that was included in the clustering analysis above was used as the reference gene set for enrichment analysis. Two thousand base pairs upstream of the first transcript in each Worm Base entry for the IIS-regulated genes in our study were searched for the forward and reverse sequences of the DAE and DBE as reported in reference 35.

**Data availability.** All raw sequencing data analyzed in this study have been uploaded to the NCBI Sequence Read Archive (SRA) database under accession number PRJNA717670.

## SUPPLEMENTAL MATERIAL

Supplemental material is available online only.
**SUPPLEMENTAL FILE 1**, XLSX file, 1.7 MB.

## ACKNOWLEDGMENTS

We thank Chris Walls for maintaining Juniata's computer cluster, on which many of our analyses were run.

This work was funded by NIH grant R15AG052933 (J.P.C.) and R15AG063103 (R.L.).

V.P.B., R.L., and J.P.C. designed the study. J.D.B., C.S.E., and G.R.L. grew the *C. elegans*. L.V.U. performed RNA extractions and library preparation. C.J.B., V.P.B., N.S.H., G.R.L., and J.R.C.S. analyzed the sequence data. J.D.B., J.R.C.S., and G.R.L. wrote the paper, with input from C.J.B., V.P.B., J.P.C., and R.L.

We declare no conflicts of interest.

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
