## [Reviewer comments · Microbiology Spectrum]

Microbiology Spectrum

Temporal Transcriptomics of Gut *E. coli* in *C. elegans* Models of Aging

Joshua Brycki, Jeremy Chen See, Gillian Letson, Cade Emler, Lavinia Unverdorben, Nathan Heibeck, Colin Brislawn, Vincent Buonaccorsi, Jason Chan, and Regina Lamendella

Corresponding Author(s): Regina Lamendella, Juniata College

Review Timeline:

Submission Date:

June 17, 2021

Accepted:

July 22, 2021

Editor: Kevin Theis

Reviewer(s): The reviewers have opted to remain anonymous.

Transaction Report:

DOI: <https://doi.org/10.1128/Spectrum.00498-21>

July 22, 2021

Dr. Regina Lamendella
Juniata College
Biology
1700 Moore Street
Huntingdon, PA 16652

Re: Spectrum00498-21 (Temporal Transcriptomics of Gut *E. coli* in *C. elegans* Models of Aging)

Dear Dr. Regina Lamendella:

Your manuscript has been accepted, and I am forwarding it to the ASM Journals Department for publication. You will be notified when your proofs are ready to be viewed.

Here are a few minor edits to address moving forward:

In the legends for Supplemental Figures 3-6, shouldn't the text for panel B read "PLS-DA plot of [genotype] samples grouped by time point"?

In the legend for Table 4, consider changing first sentence to "Summary of ADONIS comparisons based on *E. coli* expression using the Bray-Curtis distance metric."

For Table 5, it is written that "The regulator UDP-3-O-(3-hydroxymyristoyl) glucosamine N-acyltransferase (LpxD) had a slope of -0.053." In the table, the slope is -0.050.

Doublecheck that taxonomic names and genotype names are italicized throughout the manuscript.

In the Methods, in the sentence ""Pathway networks were visualized for the three differentially expressed three biofilm pathways identified by the LEfSe analysis for time point (Figure 3)," should one of the "three"s be removed?

Sincerely,

Kevin Theis

Journals Department
Supplemental Figures 1-22: Accept
Supplemental Tables 1-40: Accept